# STUTTER MAKES LARGE LANGUAGE MODELS SMARTER

## ABSTRACT

Large language models (LLMs) have achieved remarkable success in generating coherent and contextually relevant text. However, their large parameters and high memory requirements limit their efficiency and adoption in industry and academia. Recent studies have shown that dynamically adjusting inference operations can improve model performance without significantly increasing size. In this paper, we introduce the stutter mechanism, a novel method that enhances transformer models by selectively applying additional layers to more challenging tokens. This approach mimics a human speaker's stutter, allocating more computational effort where needed, thus improving language capabilities without generating excessive tokens. Our experiments with various Pythia models demonstrate that the stutter mechanism consistently enhances performance across benchmark datasets. Specifically, the Pythia-410M model, enhanced by our method, outperforms the larger Pythia-1B model on WinoGrande and WSC. Additionally, our method is data-efficient, requiring only less than 1% of the pretraining data for the additional training. These results highlight the stutter mechanism's potential to enhance LLMs' efficiency and performance in real-world applications.

## 1 INTRODUCTION

Decoder-only transformers (Radford et al., 2019; Brown et al., 2020) have become the standard for large language models. These models, such as GPT-3 and its successors, have demonstrated remarkable capabilities in generating coherent and contextually relevant text across a wide range of applications, from natural language understanding to creative writing. The architecture's simplicity, combined with its ability to scale effectively with increased data and computational resources, has made it the go-to choice for developing state-of-the-art language models. However, despite their success, there remains significant room for improvement, particularly in efficiency and adaptability to varying input complexities.

Typically, a transformer processes all inputs with the same procedure. This uniform approach, while straightforward, does not account for the varying difficulty levels of different inputs or the specific quality requirements of the output. There have been a number of attempts aiming to dynamically adapt the operation flow of a transformer to the difficulty level of the input or the requirements on the quality of the output (Snell et al., 2024). These methods include techniques such as adaptive computation time, where the model decides how many layers to apply based on the input's complexity, and dynamic layer skipping, which allows the model to bypass certain layers when they are deemed unnecessary. Such approaches aim to make the model more efficient by allocating computational resources more judiciously, thereby improving both speed and performance.

Inspired by recent upscaling studies (Kim et al., 2024; Chowdhery et al., 2023), which have shown that larger models tend to perform better across a variety of tasks, we sought to explore ways to enhance the language capabilities of existing transformer models without significantly increasing their size. Upscaling studies have demonstrated that increasing the number of parameters and layers in a model can lead to substantial improvements in performance. However, this comes at the cost of increased memory requirements and training time. Our goal was to find a method that could leverage the benefits of upscaling while mitigating its drawbacks, particularly in terms of efficiency and memory consumption. While using pause tokens (Goyal et al., 2024) is an intuitive method that allows encoding the prefix in parallel, they require substantial computational resources and time

for pretraining on the entire C4 dataset. Similarly, the work ALBERT (Lan et al., 2020) focuses on reducing memory usage by factorizing embedding parameters and sharing parameters across layers. Our approach, however, aims to improve performance through the stutter mechanism, which enhances the model's ability to utilize prior information effectively.

In this paper, we propose the stutter mechanism, a minimally intrusive method to dynamically raise the language ability of an existing transformer when needed. Our approach is based on the hypothesis that not all tokens are equally easy to generate; for at least some of them, a transformer can do better by "giving more thought" to an in-flight token by "transforming" it with more operations. It works as if a human speaker stutters when encountering a key diverging point in a speech. This analogy captures the essence of our method: by selectively applying additional layers to more challenging tokens, the model can allocate more computational effort where it is most needed, thereby improving overall performance without a significant increase in resource usage.

This paper is about, once identified, how to apply more layers. It is compatible with any methods that determine the tokens that deserve to be given more thoughts. This method is minimally intrusive, requiring only minor modifications to the existing transformer architecture, and is highly effective in enhancing the model's language capabilities.

We implemented our method on Pythia-160M, Pythia-410M, and Pythia-1B. Our experiment results show that the proposed methods effectively raised the accuracies of the Pythia models on the LAM-BADA (OpenAI), PIQA, WinoGrande, WSC, ARC-e, ARC-c, SciQ and LogiQA benchmarks. With the help of the stutter mechanism, a smaller model can even outperform a much larger model.

Our contributions are threefold:

- **Innovative Mechanism for Enhanced Language Capability**: We introduce the stutter mechanism, a novel and minimally intrusive method that dynamically allocates additional computational resources to more challenging tokens. By leveraging specific transformer layers to serve as a silent thinking process, our approach improves the model's language capabilities without significantly increasing resource usage. This mechanism is compatible with existing methods for identifying tokens that require more computational effort, making it a versatile addition to current transformer architectures.

- **Performance Improvements on Various Benchmarks**: We demonstrate that the stutter mechanism significantly enhances the performance of transformer models on various benchmarks. Specifically, our experiments show that the Pythia-410M model, enhanced by the stutter mechanism, outperforms the larger Pythia-1B model on WinoGrande and WSC. These results highlight the practical effectiveness of our approach in real-world applications.

- **Data and Computational Efficiency**: We show that only one billion tokens (less than 1% of the pretraining data) are sufficient to train the stutter mechanism, reducing the computation time and cost by a significant amount. Therefore, our method is not only effective but also practical for large-scale deployment.

## 2 BACKGROUNDS

In this section, an overview of key concepts and techniques relevant to the development of transformer models is provided. We discuss the architecture and scaling trends of decoder-only transformers, methods for upscaling and pruning, and approaches to improve computational efficiency. Additionally, we explore the loss functions used in training and the confidence levels of transformers in token prediction.

### 2.1 DECODER-ONLY TRANSFORMERS

The Generative Pre-trained Transformer (GPT) series by OpenAI showcases the power of decoder-only transformer architectures (Radford et al., 2019; Brown et al., 2020). GPT-2, released in 2019 with 1.5 billion parameters, demonstrated impressive text generation capabilities. GPT-3, introduced in 2020, expanded to 175 billion parameters, significantly enhancing performance and enabling more complex and accurate text generation. This progression highlights the trend that increasing model parameters leads to substantial performance improvements (Hoffmann et al., 2022).

As the number of parameters increases, the depth of the model also tends to increase. For example, GPT-2 has 48 layers, while GPT-3 scales up to 96 layers. This trend is also observed in various large language models where more layers are added to accommodate the growing number of parameters, thereby enhancing the model's capacity to learn complex patterns and dependencies in the data (Zhao et al., 2023). This scaling law is further supported by studies showing that larger models continue to improve performance with increased size (Kaplan et al., 2020).

## 2.2 UPSCALING

While increasing the number of parameters and layers can enhance model performance, it also introduces significant computational challenges. To address these challenges, upscaling methods are employed to increase the parameter count and the depth of a transformer. These methods can be broadly categorized into training-free attempts and upscale-and-train attempts. Training-free upscaling involves techniques such as parameter sharing and repeating layers without additional training. Recently, merged LLMs have shown success in improving performance without re-training. An evolutionary algorithm is proposed in (Akiba et al., 2024) to search for a better merge combination which is costly and limits the number of repetitions.

On the other hand, upscale-and-train methods involve increasing the model size and then training it on large datasets to achieve better performance. For instance, the SOLAR 10.7B model demonstrates effective depth upscaling techniques that significantly enhance model performance (Kim et al., 2024). Additionally, the authors in (Chowdhery et al., 2023) discuss how scaling pathways can be used to efficiently upscale models.

## 2.3 LAYERS SKIPPING AND PRUNING

Despite the benefits of upscaling, the increased model size can lead to inefficiencies during inference. To decrease the runtime computational requirements of a transformer, various methods such as layer skipping and pruning are employed. Layer skipping involves dynamically skipping certain layers during inference based on the input data, thereby reducing the computational load. Pruning, on the other hand, involves removing less important weights or neurons from the model, which can significantly reduce the model size and inference time while conceding some performance. The authors in (Fan et al., 2024) explore these techniques in detail, showing how selective layer usage can maintain performance while reducing computational costs. Another approach proposed in (Liu et al., 2023; Li et al., 2022) demonstrates that layer sparsity can be contextualized, suggesting that not all layers are necessary for processing simpler input tokens. In addition, observations from (Halawi et al., 2023) show that early-exiting in critical layers (around layer 28 in GPT2-XL) improves the model performance.

## 2.4 HOW CONFIDENT IS A TRANSFORMER ON A GIVEN TOKEN

Understanding the training and inference processes is essential (Lieberum et al., 2024), but it is equally important to evaluate the model's confidence in its predictions. The confidence of a transformer on a given token can be measured by the probability distribution it outputs for the next token prediction. Studies have shown that transformers can generate high-confidence predictions for certain tokens, which can be used to gauge the model's certainty in its predictions. While there are extensive studies on the overall performance of transformers in generating sequences, there is ongoing research to understand the confidence levels at the token level. For example, authors in (Sun et al., 2024; Lad et al., 2024) discuss the confidence and interpretability of transformer layers in generating specific tokens. Additionally, the study delves into how models process and generate tokens with varying levels of confidence (Halawi et al., 2023).

# 3 METHODS

## 3.1 ARCHITECTURE

In a prototypical transformer with $L$ layers and a sequence of tokens $X = \{x_1, ..., x_N\}$, we denote the input representation of layer $l$ and token $n$ as $h_l^n$. The input token $n$ to the first layer is

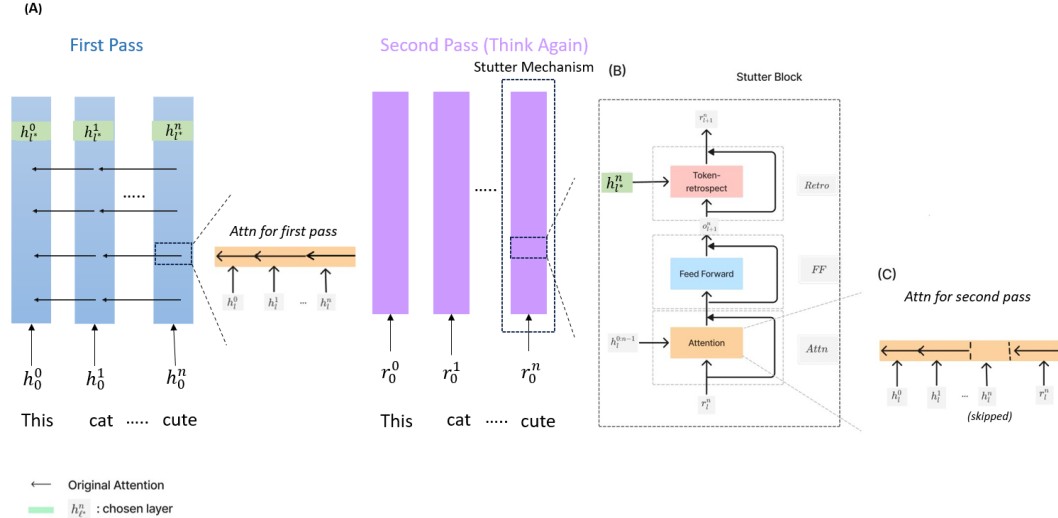

Figure 1: Overview of the proposed model architecture and stutter mechanism. (Here, we illustrated stuttering at every token. In practice, we don't need to stutter at every token.) (A) **Model Architecture**. Each blue column represents and inference step at the first pass, and each purple column represents and inference step at the second pass. For the first pass, we input the whole sequence (e.g. This cat ... cute), tokens are embedded as $h_0^0$, $h_0^1$ ... $h_0^n$ and propagated through the base model to store the hidden states $h_{l*}^0$, $h_{l*}^1$, ..., $h_{l*}^n$. For standard token (e.g. $h_0^0$), it does standard self-attention with $h_0^{0:n-1}$ and will skip all hidden states corresponding to previous stutter tokens. For the second pass, the stutter mechanism is applied. The same sequence is fed into the model again for the second pass, embedded as $r_0^0 = h_0^0$, $r_0^1 = h_0^1$, ... ,$r_0^n = h_0^n$, which is the same as the first pass input embedding. For stutter token in second pass (e.g. $r_0^n$), the self-attention part will skip hidden states corresponding to both $h_0^n$ and all previous stutter tokens $r_0^{0:n-1}$. (B) **Stutter block.** In the second pass, each layer includes a stutter block. Our proposed token-retrospect map, which is for collecting information stored in the hidden states of the chosen layer ($h_{l*}^n$), is applied after the pretrained feed-forward and attention mechanisms, along with a residual connection. (C) **Skipped attention**. As illustrated, during the second pass, the attention mechanism skips the hidden state from the first pass while still attending to the previous tokens as usual.

represented as $h_0^n$, corresponding to the embedding of the previous output token. As the token progresses through the layers of the transformer, the transformation applied by layer $l$ is described by the equation $h_{l+1}^n = \text{FF}(\text{Attn}(h_l^{0:n-1}, h_l^n))$. Here, FF represents the feed-forward network, Attn denotes the attention mechanism, and $h_l^{0:n-1}$ represents the representation of all previous tokens in the corresponding layer [1]. By the end of $L$ layers, the output of the last layer $h_{L+1}^n$ is converted into the logits of tokens by language head $y^n = \text{Head}(h_{L+1}^n)$.

### 3.1.1 STUTTER MECHANISM

The stutter mechanism is designed to enhance the model's ability to process and understand a specific token $n$ by performing the inference for that token twice. This approach allows the model to "think again" about the token, leveraging additional semantic information gathered during the first pass.

As illustrated in Figure 1, in the first pass, the model processes the token $n$ by doing standard self-attention with token $0 : n - 1$ and stores the hidden state $h_{l*}^n$, which captures the semantic information of the token. Research indicates that the last layer of a transformer model behaves differently from other layers (Lad et al., 2024; Liu et al., 2023), often filtering out a lot of information

---
[1]For simplicity, we have omitted the notation for positional embedding, normalization layers, and residual connections, although they are typically present in transformer architectures

and focusing primarily on the current output. Therefore, the hidden state before the last layer $h_{l*}^n = h_L^n$ is chosen as the semantic information from the first pass.

In the second pass, the stutter mechanism is applied, and each layer includes a stutter block. This block consists of the original pretrained attention (Attn) and feed-forward (FF) components, along with the newly introduced token-retrospect map. The token-retrospect map utilizes the information stored in $h_{l*}^n$ from the first pass. The input to the first layer in the second pass, $r_0^n = h_0^n$, is the same as in the first pass. The intermediate representation of layer $l$ in the second pass is denoted as $r_l^n$, which performs self-attention with $h_l^{0:n-1}$, skipping both $h_l^n$ and $r_l^{0:n-1}$.

During the "think again" phase (i.e., the second pass), the input $r_l^n$ of the layer $l$ first goes through the original architecture, producing an output $o_{l+1}^n$:

$$o_{l+1}^n = \text{FF}(\text{Attn}(h_l^{0:n-1}, r_l^n))$$

The result $o_{l+1}^n$ is then integrated with the hidden states from the first pass $h_{l*}^n$ using the token-retrospect map. For layers $l$ not higher than the chosen layer $l^*$, the transformation is described by:

$$r_{l+1}^n = \text{token-retrospect}(o_{l+1}^n, h_{l*}^n) + o_{l+1}^n$$

Here, $o_{l+1}^n$ represents the residual connection. The details of the proposed token-retrospect map are given in the subsection below.

### 3.1.2 TOKEN-RETROSPECT MAP

The token-retrospect map is the key component of the stutter mechanism. It is defined as:

$$\text{token-retrospect}(o_{l+1}^n, h_{l*}^n) = \left( \frac{q_{o_{l+1}^n}^T k_{h_{l*}^n}}{\sqrt{d_k}} \right) v_{h_{l*}^n} \qquad \forall l \leq l^*,$$

where $q_{o_{l+1}^n} = W_l^q o_{l+1}^n$, $k_{h_{l*}^n} = W_l^k h_{l*}^n$, $v_{h_{l*}^n} = W_l^v h_{l*}^n$ and $W_l^q$, $W_l^k$ and $W_l^v$ are additional attention parameters for training. This map uses the attention mechanism to integrate the output of the original architecture with the semantic information from the first pass, enhancing the model's ability to "think again" and refine its understanding of the token.

### 3.2 INFORM A SELF-INSIGHT TO A TRANSFORMER

To focus on the key concept of a transformer generating a token with the help of its own insights, we adhere closely to the self-attention mechanism of the underlying transformer. In our approach, we apply attention to two hidden states linearly, without using the Softmax function. This allows the model to directly leverage the stored hidden states from a previous layer, providing additional context and insight during the token generation process. The proposed stutter mechanism integrates the result of the original model with the hidden states from the chosen layer, thereby enhancing the model's ability to generate tokens with greater context and insight.

### 3.3 TRAINING

To train the proposed architecture, we start with an existing transformer and freeze all its weights except those in the token-retrospect map. Our primary objective is to demonstrate the effectiveness of the stutter mechanism, so the selection of specific tokens to stutter is beyond the scope of this paper. Therefore, during training, we stutter every token exactly once.

For the training set preparation, we perform a pass of the training sequence $X$ through the inherited transformer to capture $h_{l*}^{0:N}$. Following this, we train the stutter transformer by stuttering at every token. In the new training model, each layer is augmented by attending to the additional input. Specifically, additional parameters in the token-retrospect map are introduced for this purpose. During the training process, only the additional attention parameters are trained, while the rest of the network remains frozen.

The number of additional parameters is the same as the pretrained self-attention parameters, which constitute only 10% of the entire model. Since the number of parameters does not increase significantly, it doesn't require a large amount of data for training. In our experiment, performance

saturation was achieved with only 1 billion tokens, which is less than 1% of the pretraining data, showing competitive data efficiency.

## 3.4 LOSS

We use the next token prediction loss as our primary loss term. This loss function is essential for language modeling tasks because it evaluates the model's ability to predict the next token in a sequence given the previous tokens. The next token prediction loss is especially useful when there is no larger model with the same tokenization available.

## 4 EXPERIMENTS

### 4.1 EXPERIMENTAL SETUP

We utilized "The Pile (Gao et al., 2020)" as our training dataset, a large-scale and diverse text corpus originally segmented into 30 compressed files, each containing approximately 7 million samples. Note that all existing weights of the transformers are frozen, and only weights of the newly introduced token-retrospect map are trainable. Since the token-retrospect map has significantly fewer parameters than the original model, we randomly selected a subset of "The Pile" for training. Specifically, we used 1 billion tokens to train our token-retrospect map. Following the Pythia (Biderman et al., 2023) model's approach, we employed a parallel training setting where hidden states, MLP outputs, and attention outputs are combined. In line with this setting, we also integrated our token-retrospect outputs to further enhance the model's performance.

For our experiments, we utilized 4 NVIDIA A6000 GPUs to train the stutter mechanism. The training dataset consisted of 1 billion tokens, which were trained for 1 epoch. We employed a learning rate of 5e-5, using a cosine scheduler with a warmup ratio of 0.01. The optimizer used was Adam, and we set the gradient accumulation steps to 8.

In the initial pass of the model with $L$ layers, we store the hidden states of the $(L-1)$-th layer for each token. The token-retrospect map was initialized using Gaussian initialization with a mean of 0 and a standard deviation of 1e-5 before the stuttering process. We store our checkpoint models every 5000 steps and evaluate them on the LAMBADA (OpenAI) dataset as the in-training evaluation. Stuttering is enabled for all tokens during inference and each token is only allowed to repeat once.

### 4.2 EVALUATION

#### 4.2.1 PERFORMANCE ON VARIOUS BENCHMARK DATASETS

- **Pythia Model**: We use Pythia 160M, 410M, and 1B as our base models, showing that the proposed stutter mechanism is effective for various model scales.
- **Benchmarks**: We evaluate models on the LAMBADA (OpenAI) (Radford et al., 2019), PIQA (Bisk et al., 2020), WinoGrande(Sakaguchi et al., 2019), WSC(Levesque et al., 2012), ARC-e (Clark et al., 2018), ARC-c(Clark et al., 2018), SciQ(Welbl et al., 2017) and LogiQA(Liu et al., 2020) datasets. These datasets are designed to test various aspects of language understanding and reasoning, providing a comprehensive evaluation of the model's capabilities.

### 4.3 RESULTS

The results of our experiments demonstrate the effectiveness of the proposed architecture, particularly the stutter mechanism, which enhances the performance of the Pythia models by incorporating additional hidden states during the first pass and reprocessing the tokens. This section presents a detailed comparison of the vanilla Pythia models and the stutter Pythia models trained on 1B tokens.

#### 4.3.1 PERFORMANCE ANALYSIS OF PYTHIA MODELS

This subsection compares the performance of Pythia-160M, Pythia-410M, and Pythia-1B on various benchmarks, evaluating both vanilla and "with stutter" models. As shown in Table 1, Pythia-160M-

Table 1: Performance of Pythia-160M and Pythia-160M-Stutter on Various Benchmarks

| Benchmark | Metric (Acc) | Pythia-160M | Pythia-160M-Stutter |
|---|---|---|---|
| LAMBADA (OpenAI) | 5-shot / 0-shot | 0.271 / 0.353 | 0.295 / 0.383 |
| PIQA | 5-shot / 0-shot | 0.625 / 0.623 | 0.631 / 0.625 |
| WinoGrande | 5-shot / 0-shot | 0.513 / 0.513 | 0.519 / 0.519 |
| WSC | 5-shot / 0-shot | 0.575 / 0.575 | 0.615 / 0.615 |
| ARC-e | 5-shot / 0-shot | 0.442 / 0.436 | 0.456 / 0.449 |
| ARC-c | 5-shot / 0-shot | 0.180 / 0.194 | 0.185 / 0.180 |
| SciQ | 5-shot / 0-shot | 0.780 / 0.754 | 0.789 / 0.776 |
| LogiQA | 5-shot / 0-shot | 0.235 / 0.196 | 0.225 / 0.201 |

Table 2: Performance of Pythia-410M and Pythia-410M-Stutter on Various Benchmarks

| Benchmark | Metric (Acc) | Pythia-410M | Pythia-410M-Stutter |
|---|---|---|---|
| LAMBADA (OpenAI) | 5-shot / 0-shot | 0.442 / 0.516 | 0.449 / 0.524 |
| PIQA | 5-shot / 0-shot | 0.680 / 0.667 | 0.688 / 0.682 |
| WinoGrande | 5-shot / 0-shot | 0.533 / 0.532 | 0.538 / 0.538 |
| WSC | 5-shot / 0-shot | 0.659 / 0.659 | 0.670 / 0.670 |
| ARC-e | 5-shot / 0-shot | 0.545 / 0.518 | 0.553 / 0.519 |
| ARC-c | 5-shot / 0-shot | 0.218 / 0.214 | 0.219 / 0.219 |
| SciQ | 5-shot / 0-shot | 0.892 / 0.815 | 0.894 / 0.829 |
| LogiQA | 5-shot / 0-shot | 0.230 / 0.216 | 0.215 / 0.213 |

Stutter generally improves performance overall benchmarks compared to Pythia-160M, notably increasing LAMBADA (OpenAI) 5-shot accuracy from 0.271 to 0.295 and 0-shot accuracy from 0.353 to 0.383. For the WSC benchmark, both 5-shot and 0-shot accuracies increase from 0.575 to 0.615.

Similar results can be found in Tables 2 and 3, where Pythia-410M and Pythia-1B also benefit from the stutter mechanism. Notably, Pythia-410M-Stutter achieves performance close to Pythia-1B, and in some cases, even outperforms it. In the WSC benchmark, Pythia-410M-Stutter achieved an accuracy of 0.670 compared to Pythia-1B's 0.666 in both 5-shot and 0-shot settings. Similarly, on the WinoGrande dataset, Pythia-410M-Stutter outperformed Pythia-1B, achieving an accuracy of 0.538 versus 0.534 in both 5-shot and 0-shot settings.

On the LAMBADA (OpenAI), WinoGrande, WSC, and SciQ benchmarks, the stutter mechanism shows significant and consistent improvements regardless of model size or different model families. These results indicate that the stutter mechanism effectively enhances model performance across various contexts, demonstrating its broad applicability and stability.

Overall, the introduction of the stutter mechanism enhances performance across various benchmarks. Notably, on the LAMBADA (OpenAI), WinoGrade, WSC, and SciQ benchmarks, the stutter mechanism shows significant and consistent improvements regardless of model size or model family (Tables 1, 2, 3, 6). These results indicate that the stutter mechanism effectively enhances model performance across diverse contexts, demonstrating its broad applicability and stability. This makes the models competitive with larger counterparts without incurring significant computational costs.

### 4.3.2 CORRECTNESS TRANSITION

In order to dig into the effectiveness of the stutter mechanism, we make the statistics of the number of tokens that are improved from incorrect to correct and vice versa. In table 4, we can see that while the stutter mechanism does enable some tokens to be corrected (from wrong to right), it also introduces errors (from right to wrong). The net effect across all three models (Pythia-160M, Pythia-410M, and Pythia-1B) shows a greater number of tokens transitioning from correct to incorrect, indicating that the mechanism has an overall positive impact on the performance of the Pythia models across

Table 3: Performance of Pythia-1B and Pythia-1B-Stutter on Various Benchmarks

| Benchmark | Metric (Acc) | Pythia-1B | Pythia-1B-Stutter |
|---|---|---|---|
| LAMBADA (OpenAI) | 5-shot / 0-shot | 0.485 / 0.562 | 0.509 / 0.578 |
| PIQA | 5-shot / 0-shot | 0.714 / 0.707 | 0.716 / 0.700 |
| WinoGrande | 5-shot / 0-shot | 0.534 / 0.534 | 0.542 / 0.542 |
| WSC | 5-shot / 0-shot | 0.666 / 0.667 | 0.681 / 0.681 |
| ARC-e | 5-shot / 0-shot | 0.586 / 0.569 | 0.596 / 0.572 |
| ARC-c | 5-shot / 0-shot | 0.256 / 0.244 | 0.257 / 0.240 |
| SciQ | 5-shot / 0-shot | 0.917 / 0.839 | 0.927 / 0.853 |
| LogiQA | 5-shot / 0-shot | 0.238 / 0.225 | 0.216 / 0.224 |

Table 4: Correctness Transition Matrix on Lambada (OpenAI)

| # token | Pythia-160M | | Pythia-410M | | Pythia-1B | |
|---|---|---|---|---|---|---|
| | To Right | To Wrong | To Right | To Wrong | To Right | To Wrong |
| From Right | 1681 | 135 | 2517 | 143 | 2766 | 131 |
| From Wrong | 297 | 3040 | 188 | 2305 | 214 | 2042 |

different sizes. Also, the proportion of the improving tokens is roughly 3-5%, decreasing as the baseline model performs better.

### 4.3.3 KL DIVERGENCE ANALYSIS

To verify if the improved small model is more aligned with the model with larger sizes, we evaluate the KL divergence of Pythia-160M and Pythia-160M-Stutter with a larger model. Taking Pythia-1B as the target distribution, the figure shows the averaged token-wise KL divergence between Pythia-160M-Stutter and Pythia-1B is smaller than that between Pythia-160M and Pythia-1B. This indicates that the stutter mechanism effectively aligns the output distribution of the smaller Pythia-160M model closer to that of the larger Pythia-1B model. Notably, there are a few exceptions (e.g., WinoGrande and WSC) where the KL divergence slightly increases. As we discussed in Section 4.3.1, the performance is also improved over these two datasets. That means our mechanism improves the performance in the way orthogonal to upscaling model sizes. Since both datasets focus the pronoun resolution and common sense reasoning, in other words, the stutter mechanism might exhibit a deep contextual understanding ability that could not be derived from increasing the number of parameters.

### 4.4 ABLATION STUDY

We conducted an ablation study to evaluate the performance of our stutter transformer under different settings. The study focused on different stutter times and the effectiveness of the chosen layer.

### 4.4.1 DIFFERENT STUTTER TIMES

In Table 5 we compare the performance of Pythia-160M, Pythia-410M, and Pythia-1B models on the LAMBADA (OpenAI) benchmark, evaluating the effects of stuttering once versus stuttering twice.

For Pythia-160M, stuttering twice slightly improves perplexity from 26.927 to 26.636 and accuracy from 0.383 to 0.387. For Pythia-410M, stuttering twice also reduces perplexity and increases accuracy. For Pythia-1B, stuttering twice slightly reduces perplexity from 7.439 to 7.403 but results in a marginal decrease in accuracy from 0.578 to 0.576.

While stuttering twice generally enhances performance, the improvements are often marginal. For instance, Pythia-410M with stutter once achieves an accuracy of 0.524, which is very close to the 0.527 accuracy with stutter twice, making the former a more cost-effective option. Therefore, given

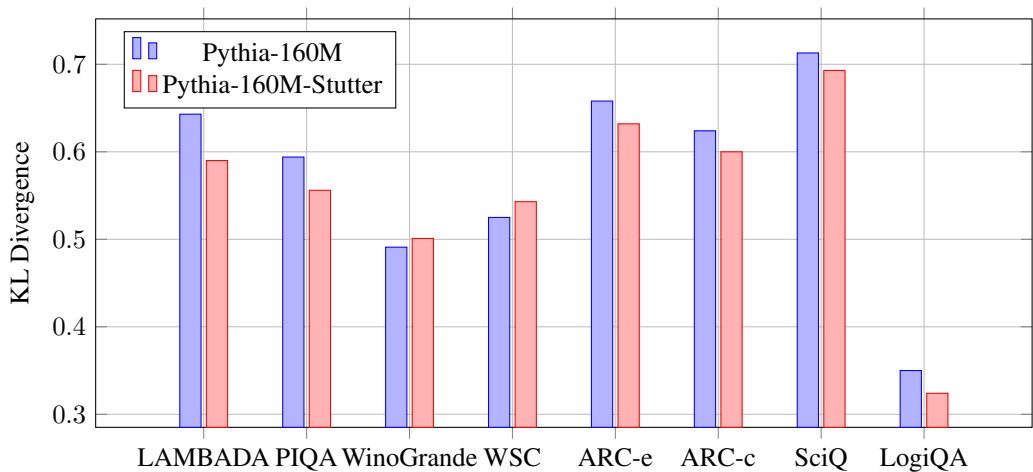

Figure 2: KL Divergence evaluation over 8 benchmarks.

Table 5: Pythia models with different stutter times on LAMBADA (OpenAI)

| Models | Metric | stutter once | stutter twice |
|---|---|---|---|
| Pythia-160M | perplexity/acc | 26.927/0.383 | 26.636/0.387 |
| Pythia-410M | perplexity/acc | 10.387/0.524 | 10.272/0.527 |
| Pythia-1B | perplexity/acc | 7.439/0.578 | 7.403/0.576 |

the additional computational cost, stuttering once is generally a more efficient strategy for optimizing model performance.

### 4.4.2 EFFECTIVENESS OF $h_{l*}$

To assess the effectiveness of the chosen layer, we experimented with employing the stutter mechanism at different layers of the Pythia-160M model:

- Layer 10: The stutter mechanism attends to the output hidden states of the $L - 2$th layer.

- Layer 11: The stutter mechanism attends to the output hidden states of the $L - 1$th layer.

- Layer 12: The stutter mechanism attends to the output hidden states of the last layer.

These experiments were designed to determine the optimal layer for capturing the intermediate insights of the transformer and to evaluate the impact of different layers on the model's performance.

As shown by our results in Figure 3, our findings suggest that attending at specific intermediate positions can indeed boost performance. While attending to layer 10 and layer 11 yields similar performance, layer 12 generally results in lower improvements across tasks except for LogiQA. As indicated by previous work (Lad et al., 2024), the last layer filters out some semantic information and might only contain the necessary information for the next token. That explains why layer 12 is generally not a good layer to attend to. Surprisingly, LogiQA, the most difficult dataset among benchmarks, is improved significantly by attending the last layer. One explanation is domain knowledge is stored in a specific layer. Another explanation is excluding some of the information is helpful in tasks like LogiQA, where all the options are quite similar and confusing. Notably, We observe consistent declines in performance in the ARC-C benchmark regardless of the attended layer. That is because this dataset contains many lengthy options. While the stutter mechanism performs well in providing short answers, the additional stutter mechanism might weaken the logits contributed by the attention mechanism, resulting in inferior performance for long-context benchmarks.

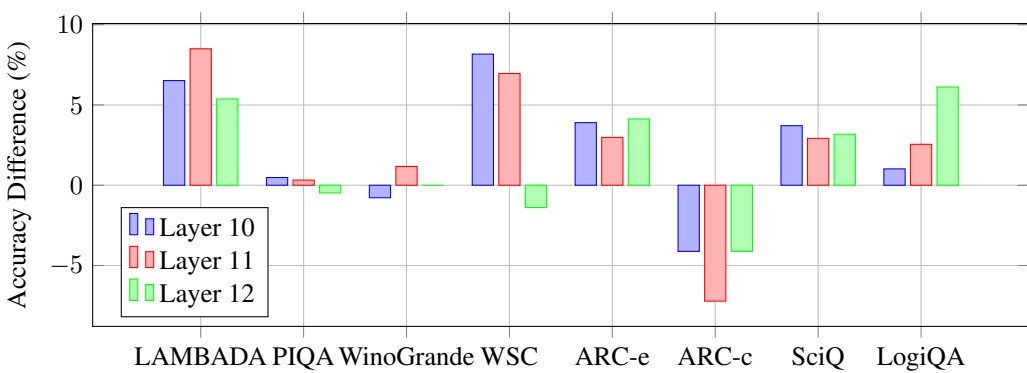

Figure 3: Pythia-160M-Stutter with Different Chosen Layers (0-shot) - Baseline Subtracted

## 5 CONCLUSION AND FUTURE WORK

We propose the stutter mechanism that effectively enhances the performance of LLMs by facilitating an extended thinking process. This approach not only addresses the limitations of increasing model sizes but also optimizes computational efficiency by tailoring the processing requirements to the complexities of different tasks. Our extensive experiments with various Pythia models demonstrate that the stutter mechanism consistently improves performance across benchmark datasets. With the help of the stutter mechanism, a smaller model can even outperform a much larger model. While our proposed method has shown promising results in enhancing the language capabilities of transformer models, there are several avenues for future research and development that could further improve and extend our approach. Here, we outline some potential directions for future work:

- **Efficient Repeating Mechanism**: Future work could optimize the repeating mechanism by dynamically determining the exact number of layers each token needs, rather than applying the entire network. This real-time assessment would minimize unnecessary computations, enhancing efficiency and performance by adapting more precisely to input complexities.
- **Different Ways of Heuristic**: Refining heuristics for the "stuttering" mechanism is another key area. Using fine-tuning or Reinforcement Learning from Human Feedback (RLHF), we can develop smarter heuristics to decide when to stutter, how many times, and when to stop, making the model more adaptive and effective in handling reasoning tasks.
- **Interpreting Reasoning Mechanism**: Understanding how LLMs reason is crucial for building trust and transparency in AI systems. By analyzing attention distributions, we can identify which attention heads contribute most to reasoning ability. This insight can help us understand the internal mechanisms of LLMs and how they process information to arrive at conclusions. Future work could focus on developing methods to visualize and interpret these attention patterns, potentially guiding further improvements in model design.

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

## A  APPENDIX

### A.1  TIME COMPLEXITY OF STUTTER MECHANISM

**Training**: In our current setting in training, we stutter at every token. Our method employs a two-pass process. For a given sample $X = x_1, x_2, \ldots, x_n$:

1. **First Pass**: We store the hidden states $h_{l*}^1, h_{l*}^2, \ldots, h_{l*}^n$ for each token in the sequence.

2. **Second Pass (Stutter Phase)**: During this phase, we utilize the stored hidden states by passing both the stored hidden states and the current token to the model. For instance, to generate the final prediction for the next token of $x_3$, we input $x_2$ and $h_{l*}^2$ to the stutter model. The original Feed-Forward (FF) and Attention (Attn) mechanisms perform the same as in the base model, where $x_2$ will attend on $x_1$. Our token retrospect mechanism performs linear attention using $x_2$ and $h_{l*}^2$ to extract information from the first pass to generate the next token.

If the base model has time complexity $O(n)$, the time complexity of the stutter model is $2 \times O(n)$ since we run each sample twice, remaining the same time complexity as the base model.

**Inference**: In the benchmark dataset we use for inference, the only token we use for stutter is the choice of the samples. In the Lambada-openai dataset, it only predicts the last token. In the multiple-choice dataset, the only tokens we stutter are the choice tokens. Two examples from each type of dataset is given below:

Table 6: Performance of Llama-1B and Llama-1B-Stutter on Various Benchmarks

| Benchmark | Metric (Acc) | Llama-1B | Llama-1B-Stutter |
|---|---|---|---|
| Lambda-openai | 5-shot / 0-shot | 0.571 / 0.622 | 0.588 / 0.624 |
| PIQA | 5-shot / 0-shot | 0.750 / 0.744 | 0.751 / 0.751 |
| Winograde | 5-shot / 0-shot | 0.599 / 0.599 | 0.599 / 0.599 |
| WSC | 5-shot / 0-shot | 0.751 / 0.751 | 0.762 / 0.762 |
| ARC-Easy | 5-shot / 0-shot | 0.697 / 0.655 | 0.697 / 0.649 |
| ARC-Challenge | 5-shot / 0-shot | 0.349 / 0.317 | 0.347 / 0.326 |
| SciQA | 5-shot / 0-shot | 0.952 / 0.912 | 0.958 / 0.919 |
| LogiQA | 5-shot / 0-shot | 0.209 / 0.209 | 0.220 / 0.226 |

1. For example, in the Lambada dataset, given the context:

   He heard Rihanna speak 'The Queen wants you in her carriage.' Tom spoke 'No, I'm not going in some asylum.' Ran was seen standing next to him spoke 'It's just for a private talk with you that's all.' Tom groaned and went inside the carriage to sit down next to the

   The goal is to predict "Queen". So we actually just stutter at the last token.

2. For another example from a multiple-choice dataset:

   Question: "George wants to warm his hands quickly by rubbing them. Which skin surface will produce the most heat?"
   Choice: ["dry palms", "wet palms", "palms covered with oil", "palms covered with lotion"]

   The test was to concatenate the question and each answer, so there will be four inputs, and the model calculation is given by the sum of the lowest log probability of the choice part:

   (a) "George wants to warm his hands quickly by rubbing them. Which skin surface will produce the most heat? dry palms"

   (b) "George wants to warm his hands quickly by rubbing them. Which skin surface will produce the most heat? wet palms"

   (c) "George wants to warm his hands quickly by rubbing them. Which skin surface will produce the most heat? palms covered with oil"

   (d) "George wants to warm his hands quickly by rubbing them. Which skin surface will produce the most heat? palms covered with lotion"

For inference, we will only stutter the choice part (which is "dry palms", "wet palms", "palms covered with oil", and "palms covered with lotion") or the last token. Therefore, the time complexity for inference with the stutter mechanism is $O(n)$ for the first pass and $O(1)$ for each stuttered token in the second pass. The overall time complexity remains $O(n)$, similar to the base model. This ensures that the stutter mechanism does not significantly increase the computational complexity during inference.

## A.2 EXPERIMENTS ON LLAMA ARCHITECTURE

To evaluate our approach on prominent models, we applied the stutter mechanism to the LLAMA 1B model and tested it on the same benchmarks reported in our paper under 0-shot and 5-shot settings. Despite training on only 125M tokens from the Pile dataset, the Llama-1B-Stutter model showed improvements across almost all benchmark datasets. As shown in Table 6, the Llama-1B-Stutter model demonstrates consistent improvements in several benchmarks, indicating the effectiveness of our approach. These results highlight the potential of the stutter mechanism to enhance the performance of state-of-the-art LLMs, even with a relatively small amount of training data.

Table 7: Performance of Stutter Models trained with 2B tokens on Various Benchmark

| Benchmark | Metric (Acc) | Pythia-160M-Stutter | Pythia-410M-Stutter | Pythia-1B-Stutter |
|---|---|---|---|---|
| Lambda-openai | 0-shot, Acc | 0.387 | 0.527 | 0.577 |
| PIQA | 0-shot, Acc | 0.624 | 0.680 | 0.702 |
| Winograde | 0-shot, Acc | 0.519 | 0.537 | 0.542 |
| WSC | 0-shot, Acc | 0.619 | 0.667 | 0.681 |
| ARC-Easy | 0-shot, Acc | 0.454 | 0.524 | 0.565 |
| ARC-Challenge | 0-shot, Acc | 0.186 | 0.220 | 0.242 |
| SciQA | 0-shot, Acc | 0.780 | 0.828 | 0.856 |
| LogiQA | 0-shot, Acc | 0.192 | 0.209 | 0.230 |

Table 8: Performance of Pythia-410M, Pythia-410M-Finetune, and Pythia-410M-Stutter on Various Benchmarks

| Benchmark | Metric (Acc) | Pythia-410M | Pythia-410M-Finetune | Pythia-410M-Stutter |
|---|---|---|---|---|
| Lambda-openai | 0-shot, Acc | 0.442 | 0.497 | 0.449 |
| PIQA | 0-shot, Acc | 0.681 | 0.677 | 0.689 |
| Winograde | 0-shot, Acc | 0.533 | 0.541 | 0.538 |
| WSC | 0-shot, Acc | 0.659 | 0.630 | 0.670 |
| ARC-Easy | 0-shot, Acc | 0.545 | 0.525 | 0.554 |
| ARC-Challenge | 0-shot, Acc | 0.218 | 0.223 | 0.219 |
| SciQA | 0-shot, Acc | 0.892 | 0.828 | 0.894 |
| LogiQA | 0-shot, Acc | 0.230 | 0.198 | 0.215 |

## A.3 CONTINUAL TRAINING FOR STUTTER MODELS

When fine-tuning large language models, it is common practice to use around 1 billion tokens. In our approach, we freeze the original model weights and only train the token-retrospect part, which consists of about 10% of the original parameters. This strategy allows us to efficiently fine-tune the model without the need for extensive computational resources.

To further investigate the impact of training on larger datasets, we conducted a series of experiments where we continually trained the token-retrospect map on increasing amounts of data. The training remained stable over extended periods and larger datasets, but the efficiency of additional training diminished. This indicates that our method is robust and does not require excessive amounts of data to achieve optimal performance.

The results of these experiments are summarized in Table 7, which shows the performance metrics for models trained on 1 billion and 2 billion tokens. As shown, the improvements from training on 2 billion tokens are minimal, reinforcing the idea that our approach achieves a good balance between efficiency and performance with 1 billion tokens.

In conclusion, while the stutter mechanism can benefit from additional training data, the marginal gains observed beyond 1 billion tokens suggest that our method is efficient and effective with a relatively modest amount of training data. This efficiency makes our approach particularly suitable for scenarios where computational resources are limited.

## A.4 FINE-TUNING AND STUTTER MECHANISM COMPARISON FOR PYTHIA-410M

To demonstrate the effectiveness of the stutter mechanism, we conducted an experiment with the Pythia-410M model, where we fine-tuned the Pythia-410M base model on the same 1 billion tokens used for training the stutter mechanism.

For training cost, fine-tuning the base Pythia-410M model on the 1 billion tokens involves updating all parameters, which incurs a significantly higher training cost compared to the Pythia-410M-Stutter model. The stutter model, in contrast, only trains approximately 10% of the full parameters, making it much more efficient.

The results of these experiments are summarized in Table 8. When comparing the Pythia-410M-Finetune model to the base Pythia-410M model, the base model outperforms the fine-tuned model on 5 benchmark datasets (PIQA, WSC, ARC-Easy, SciQA, LogiQA). When comparing the Pythia-410M-Stutter model to the Pythia-410M-Finetune model, the stutter model outperforms the fine-tuned model on 5 benchmark datasets (PIQA, WSC, ARC-Easy, SciQA, LogiQA).

These results demonstrate that the stutter mechanism is not only more efficient in terms of training cost but also competitive in performance, often surpassing the fine-tuned model on several benchmarks.

