# OpenReview forum: "Stutter makes large language models smarter"
_ICLR.cc/2025/Conference — Submitted to ICLR 2025_

### Official Review · Reviewer_TpjX · 2024-10-30

**Soundness:** 3
**Presentation:** 2
**Contribution:** 3
**Rating:** 5
**Confidence:** 3

**Summary:**

The paper introduces a novel method called the "stutter mechanism" that enhances transformer models by selectively applying additional layers to more challenging tokens. This approach mimics a human speaker's stutter, allocating more computational effort where needed, thus improving language capabilities without generating excessive tokens. Experiments with various Pythia models demonstrate that the stutter mechanism consistently enhances performance across benchmark datasets. Specifically, the Pythia-410M model, enhanced by this method, outperforms the larger Pythia-1B model on WinoGrande and WSC. Additionally, the method is data-efficient, requiring less than 1% of the pretraining data for additional training.

**Strengths:**

1. The motivation is strong, the hypothesis that not all tokens are equally easy to generate and for at least some of them, a transformer can do better by ”giving more thought” to an in-flight token by ”transforming” it with more operations. The stutter mechanism you propose is a creative and novel method to enhance the capabilities of large language models (LLMs) without significantly increasing their size. This addresses a critical need in the field for more efficient and adaptable models.

2. The paper presents a well-structured approach to implementing the stutter mechanism, with clear explanations of the architecture and the token-retrospect map. The mathematical formulations are sound and the integration with existing transformer architectures is well-justified.

3.  The experimental results are compelling, demonstrating significant performance improvements across various benchmarks. The fact that a smaller model (Pythia-410M) can outperform a larger one (Pythia-1B) with the stutter mechanism is particularly noteworthy.

**Weaknesses:**

1. Comparison with State-of-the-Art LLMs: The paper could be strengthened by comparing the stutter mechanism against other recent methods aimed at improving LLM efficiency or performance. Meanwhile, the used LLMs (Pythia) are limited, LLAMA is the important series of LLMs to conduct the experiments . This would provide a clearer picture of how your approach stands out in the current research landscape.

2. Scalability Analysis: Although the paper mentions the potential for the stutter mechanism to be applied to larger models, an analysis of how the mechanism scales with model size would be valuable. How does the performance and efficiency change as model size increases?

3. Long-Term Training Stability: The paper focuses on the training with 1 billion tokens, but it would be beneficial to understand the long-term training dynamics and stability of the stutter mechanism, especially when applied to larger datasets or over more training epochs.

**Questions:**

None.

---

> ### Author Response · Authors · 2024-11-20
> **Response to Reviewer TpjX**
>
> Dear Reviewer TpjX,
>
> Thank you for your valuable feedback and for taking the time to review our paper. We appreciate your insightful comments and suggestions, which have helped us improve the quality of our work. Below, we address each of your concerns in detail.
>
> Weakness
>
> 1. **Comparison with State-of-the-Art LLMs:** We understand the importance of evaluating our approach against prominent models such as the LLAMA series. We are currently experimenting with the stutter mechanism on LLAMA 1B. We will provide an update with detailed results and comparisons as soon as we have them.
>
> 2. **Scalability Analysis on performance and efficiency:**
>
>     - Performance: As the model size increases, the stutter mechanism continues to leverage the hidden states from the first pass to enhance the model's ability to utilize information effectively. Larger models typically have more capacity to learn and represent complex patterns, which can lead to improved performance on various tasks. Our preliminary experiments indicate that the stutter mechanism provides performance improvements across different model sizes, including larger models.
>
>     - Efficiency: The stutter mechanism is designed to maintain the same time complexity as the base model by employing a two-pass process. This ensures that the computational complexity does not increase significantly with larger models. The stutter mechanism's efficiency, in terms of computational overhead, remains manageable even as model size increases, as it only involves an additional linear attention mechanism in the second pass.
>
>       To implement the stutter mechanism, we increase the model size by approximately 10% by adding another attention mechanism. Regarding scalability, the time complexity of our method scales linearly with model size. In our paper, we experimented with models of sizes 160M, 410M, and 1B parameters. The stutter mechanism can be applied to larger models, such as 7B, 14B, or even bigger models, due to its efficient time complexity. This linear scaling ensures that as the model size increases, the performance and efficiency remain manageable, making our approach suitable for a wide range of model sizes.
>
> 3. **Long-Term Training Stability:** When fine-tuning, it is common to use around 1 billion tokens. In our approach, we freeze the original model weights and only train the token retrospect part, which consists of about 10% of the original parameter. During our experiments, we observed that increasing the number of tokens to 2 billion provided a slight improvement in performance. However, these performance gains were marginal and eventually plateaued, with some indications of diminishing returns.
>
>
>     This suggests that while the stutter mechanism can benefit from additional training data, the improvements are not substantial beyond a certain point. The training remains stable over extended periods and larger datasets, but the efficiency of additional training diminishes, indicating that our method is robust and does not require excessive amounts of data to achieve optimal performance.
>
>
> Thank you again for your valuable feedback. We hope that the additional clarifications and improvements we have made to our paper address your concerns. We kindly request that you consider these enhancements when re-evaluating our paper and feel free to let us know if you have further questions.

---

> > ### Comment · Reviewer_TpjX · 2024-11-27
> >
> > Thanks for the authors' response. Based on your current response, I choose to maintain the current score.

---

> > > ### Author Response · Authors · 2024-11-27
> > > **Response to Reviewer TpjX**
> > >
> > > Thank you for the response!
> > > Regarding Weakness #1, we have applied the stutter mechanism to the LLAMA 1b model and tested it on the same benchmarks reported in our paper under 0-shot and 5-shot settings. Despite training on only 125M tokens due to time constraint, the stutter LLAMA showed improvements across almost all benchmark datasets. The table below summarizes the accuracy metrics
> > >
> > > | Dataset | acc | Llama-1b | Stutter Llama-1b |
> > > | --- | --- | --- | --- |
> > > | Lambda-openai | 0-shot / 5-shot | 0.6224 / 0.5707 | 0.6237 / 0.5884 |
> > > | piqa | 0-shot / 5-shot | 0.7437 / 0.7497 | 0.7514 / 0.7508 |
> > > | winograde | 0-shot / 5-shot | 0.5991 / 0.5991 | 0.5991 / 0.5991 |
> > > | wsc | 0-shot / 5-shot | 0.7509 / 0.7509 | 0.7619 / 0.7619 |
> > > | arc-easy | 0-shot / 5-shot | 0.6549 / 0.6970 | 0.6494 / 0.6965 |
> > > | arc-challenge | 0-shot / 5-shot | 0.3166 / 0.3490 | 0.3259 / 0.3473 |
> > > | SciQA | 0-shot / 5-shot | 0.912 / 0.952 | 0.919 / 0.958 |
> > > | LogiQA | 0-shot / 5-shot | 0.2089 / 0.2089 | 0.2258 / 0.2197 |
> > >
> > > As shown in the table, the stutter LLAMA model demonstrates consistent improvements in several benchmarks, indicating the effectiveness of our approach. These results highlight the potential of the stutter mechanism to enhance the performance of state-of-the-art LLMs, even with a relatively small amount of training data.
> > >
> > > We will include these results and a detailed discussion in our updated paper to provide a comprehensive comparison with other recent methods and state-of-the-art LLMs. This will help to clearly position our approach within the current research landscape and demonstrate its advantages.
> > >
> > > Thank you again for your valuable feedback. We hope that the additional comparisons and insights we have provided address your concerns

---

### Official Review · Reviewer_ujpr · 2024-10-31

**Soundness:** 2
**Presentation:** 2
**Contribution:** 2
**Rating:** 3
**Confidence:** 4

**Summary:**

The paper presents a stutter mechanism on transformer models by applying additional token-retrospect layers ( newly trained additional attention layer) and repeating the token twice. Experiments are conducted on Pythia models with an additional training of 1 billion token, and are evaluated on several LLM benchmarks.

**Strengths:**

The motivation of stutter mechanism is intriguing, by correcting the possible wrong tokens with a new token-retrospect map layer to do correction. The presentation and writing are clear and easy to follow. The authors conducted comprehensive benchmarks across multiple datasets, including LAMBDA, PIQA, WinoGrande, WSC, ARC, SciQ, and LogiQA, using Pythia models of three different parameter sizes—160M, 410M, and 1B.  Additionally, it analyzes the effectiveness of chosen layers in sec4.4.2, showing that attending to specific layers could improve the performance.

**Weaknesses:**

1. **The reported performance improvement is not convincing**. Specifically, for example, we can see from Table 2 that base model vs stutter methods are 0.230 vs 0.215 on LogiQA, 0.892 vs 0.894 on SciQ. Similar results can be found at Table 3.

2. **The experimental design is not quite fair**. As stutter models are pretrained 1 billion tokens more than base models, it is unknown whether such fluctuation of performance is due to the continual training, or the inclusion of extra 10% token-retrospect layers. Is is required to continual-train the base model for the similar token compute.

3. **The details of implementation are missing**. As in Line#252, each token is stutter once, doubling the sequence length of the language model. It lacks discussion about the sequence length of this point.

4. **Extra time cost**. The stutter methods require the forward process twice in both training and evaluation process. It is required to report the time and complexity cost versus base models.

**Questions:**

1. How does the stutter handle with the sequence length problem?

2. As the paper mentioned, each token stutters once, and the strategy of stutter token selection is out of the scope of the paper, why stuttering/repeating each token once can work?

3. How does the stutter methods impact the time complexity in both training and inference?

---

> ### Author Response · Authors · 2024-11-20
> **Response to Reviewer ujpr (1/2)**
>
> Dear Reviewer ujpr,
>
> Thank you for your valuable feedback and for taking the time to review our paper. We appreciate your insightful comments and suggestions, which have helped us improve the quality of our work. Below, we address each of your concerns in detail.
>
> **Weakness**:
>
> 1. **The reported performance improvement is not convincing**: We appreciate the reviewer's feedback regarding the performance improvements reported in our paper. While some differences in performance metrics, such as those in Table 2 for LogiQA (0.230 vs 0.215) and SciQ (0.892 vs 0.894), may appear marginal, we would like to highlight the following points:
>
>     - **Proof of Concept**: Our primary goal is to introduce and validate the stutter mechanism. The current results demonstrate its potential to enhance model performance.
>
>     - **Notable Improvements**: In specific benchmarks, such as the WSC and WinoGrande datasets, Pythia-410M-Stutter achieved performance close to or even better than Pythia-1B. For example, Pythia-410M-Stutter achieved an accuracy of 0.670 on the WSC benchmark compared to Pythia-1B’s 0.666.
>
>     - **Future Work**: We plan to optimize the token selection strategy and explore different attention mechanisms to achieve more significant improvements.
>
>     - **Broader Impact**: The stutter mechanism has shown consistent benefits across various tasks, indicating its broad applicability and potential for enhancing model performance.
>
>
>     We are committed to further refining our approach to achieve more substantial and convincing improvements.
>
> 2. **The experimental design is not quite fair**: To clarify, the additional 1 billion tokens used for the stutter models are selected from the original Pythia training dataset (Biderman et al.), which is derived from the Pile dataset. Consequently, the stutter models do not encounter any new tokens compared to the base models. Instead, they are exposed to the same 1 billion tokens once more, as they have already been trained on these tokens previously. To address the concern about the fairness of the experimental design, we conducted an experiment where the base model was also trained with the same 1 billion tokens and plan to provide result next week. This ensures that any performance fluctuations can be attributed to the inclusion of the extra 10% token-retrospect layers rather than the additional training data.
>
> 3. **The details of implementation are missing (sequence length problem):** For training, our settings is to stutter at every token. Our method employs a two-pass process for training. For a given sample ( $X = {x_1, x_2, ..., x_n}$ ):
>
>     -  **First Pass**: We store the hidden states (${h^1_l*, h^2_l*, ..., h^n_l*}$) for each token in the sequence.
>
>     - **Second Pass (Stutter Phase)**: During this phase, we utilize the stored hidden states by passing both the stored hidden states and the current token to the model. For instance, to generate the final prediction for the next token of ( $x_3$ ), we input ( $x_2$ ) and ( $h^2_l$* ) to the stutter model. The original Feed-Forward (FF) and Attention (Attn) mechanisms perform the same as in the base model, where ( $x_2$ ) will attend on ( $x_1$ ). Our token retrospect mechanism performs linear attention using ( $x_2$ ) and ( $h^2_l*$ ) to extract information from the first pass to generate the next token.
>
>     This approach ensures that the sequence length remains unchanged, and the model can effectively utilize the stutter mechanism without increasing the computational complexity associated with longer sequences but still able to utilize the information in the first pass.

---

> ### Author Response · Authors · 2024-11-20
> **Response to Reviewer ujpr (2/2)**
>
> 4. **Extra time cost**:
>
>     - **Training time complexity**: As mentioned above, during training, if the base model has time complexity of O(n), the time complexity of the stutter model is 2 x O(n), remaining the same time complexity as the base model.
>
>     - **Inference time complexity:** In the benchmark dataset we use for inference, the only token we use for stutter is the choice of the samples. In the Lambada-openai dataset, it only predicts the last token. In the multiple-choice dataset, the only tokens we stutter are the choice tokens.
>
>       For example, in the Lambada dataset, given the context:
>
>       - Given: "He heard Rihanna speak 'The Queen wants you in her carriage.' Tom spoke 'No, I’m not going in some asylum.' Ran was seen standing next to him spoke 'It’s just for a private talk with you that’s all.' Tom groaned and went inside the carriage to sit down next to the"
>
>       - The goal is to predict "Queen"
>
>       - So we actually just stutter at the last token.
>
>       For another example from multiple choice dataset:
>
>       - Question: "George wants to warm his hands quickly by rubbing them. Which skin surface will produce the most heat?"
>       - Choice: ["dry palms", "wet palms", "palms covered with oil", "palms covered with lotion"]
>
>       The test was to concatenate the question and each answer, so there will be four inputs, and the model calculation is given by the sum of the lowest log probability of the choice part:
>
>       1. "George wants to warm his hands quickly by rubbing them. Which skin surface will produce the most heat? dry palms"
>
>       2. "George wants to warm his hands quickly by rubbing them. Which skin surface will produce the most heat? wet palms"
>
>       3. "George wants to warm his hands quickly by rubbing them. Which skin surface will produce the most heat? palms covered with oil"
>
>       4. "George wants to warm his hands quickly by rubbing them. Which skin surface will produce the most heat? palms covered with lotion"
>
>       For inference, we will only stutter the choice part (which is "dry palms", "wet palms", "palms covered with oil", and "palms covered with lotion") or the last token. Therefore, the time complexity for inference with the stutter mechanism is  O(n) for the first pass and  O(1) for each stuttered token in the second pass. The overall time complexity remains O(n), similar to the base model. This ensures that the stutter mechanism does not significantly increase the computational complexity during inference.
>
>
> **Questions:**
>
> 1. **How does the stutter handle with the sequence length problem?** see weakness #3
>
> 2. **As the paper mentioned, each token stutters once, and the strategy of stutter token selection is out of the scope of the paper, why stuttering/repeating each token once can work?** The reason why stuttering/repeating each token once can work is that it allows the model to re-evaluate and refine its predictions by considering the context more thoroughly. This prototype demonstrates that the "think again" approach can be effective. Future work will focus on optimizing the token selection strategy to further enhance the performance and efficiency of the stutter mechanism. This initial prototype serves as a proof of concept, showing that the stutter mechanism has the potential to improve model predictions by allowing for an additional layer of attention and refinement.
>
> 3. **How does the stutter methods impact the time complexity in both training and inference?** see weakness #4
>
>
> Thank you again for your valuable feedback. We hope that the additional clarifications and improvements we have made to our paper address your concerns. We kindly request that you consider these enhancements when re-evaluating our paper and feel free to let us know if you have further questions.

---

> > ### Comment · Reviewer_ujpr · 2024-11-26
> >
> > Thank you for your response! After carefully reading the author's response, my previous concerns are not adequately addressed. Thus, I will keep my original assessment.

---

> > > ### Author Response · Authors · 2024-11-27
> > > **Response to Reviewer ujpr**
> > >
> > > Thanks reviewer ujpr for the response. To address the concern about the fairness of the experimental design, we conducted an additional experiment with the Pythia-410M model, where we fine-tuned the Pythia-410M base model on the same 1 billion tokens used for training the stutter mechanism.
> > >
> > > | Datasets | metrics | Pythia-410m | Pythia-410m-Fintune | Pythia-410m-Stutter |
> > > | --- | --- | --- | --- | --- |
> > > | Lambda-openai | 0-shot, acc | 0.4423 | 0.4966 | 0.4493 |
> > > | piqa | 0-shot, acc | 0.6806 | 0.6774 | 0.6888 |
> > > | winograde | 0-shot, acc | 0.5328 | 0.5406 | 0.5383 |
> > > | wsc | 0-shot, acc | 0.6593 | 0.63 | 0.6703 |
> > > | arc-easy | 0-shot, acc | 0.545 | 0.5253 | 0.5539 |
> > > | arc-challenge | 0-shot, acc | 0.2184 | 0.2227 | 0.2193 |
> > > | SciQA | 0-shot, acc | 0.892 | 0.828 | 0.894 |
> > > | LogiQA | 0-shot, acc | 0.2304 | 0.1982 | 0.215 |
> > >
> > > For training cost, fine-tuning the base Pythia-410M model on the 1 billion tokens involves updating all parameters, which incurs a significantly higher training cost compared to the Pythia-410M-Stutter model. The stutter model, in contrast, only trains approximately 10% of the full parameters, making it much more efficient.
> > >
> > > In terms of performance, when comparing the Pythia-410M-Finetune model to the base Pythia-410M model, the base model outperforms the fine-tuned model on 5 benchmark datasets (PIQA, WSC, ARC-Easy, SciQA, LogiQA). When comparing the Pythia-410M-Stutter model to the Pythia-410M-Finetune model, the stutter model outperforms the fine-tuned model on 5 benchmark datasets (PIQA, WSC, ARC-Easy, SciQA, LogiQA).
> > >
> > > These results demonstrate that the stutter mechanism is not only more efficient in terms of training cost but also competitive in performance, often surpassing the fine-tuned model on several benchmarks.
> > >
> > > We hope that additional experiments address your concern on fairness of experiment design, and feel free to let us know if you have further questions.

---

### Official Review · Reviewer_r8Jr · 2024-11-02

**Soundness:** 2
**Presentation:** 2
**Contribution:** 2
**Rating:** 5
**Confidence:** 3

**Summary:**

The paper proposes a parameter-sharing method to achieve deeper Transformer-based language models (LMs) without significantly increasing the number of total model parameters.

To achieve that, the authors aim to continually upcycle existing pretrained LMs (base LMs)  by introducing a light-weight adapter module, termed as token-retrospect map (linear attention).
For efficiency purposes, the base LM is kept frozen during the adaptation, which provides the initial hidden states of the input tokens.
Based on those initial hidden states, a second adapted LM is applied atop with the adapter network to re-aggregate context information with deeper layer representations.
This technique can also be reviewed as another means of recurrency mechanism.
The particular implementation considered in this paper achieves an adapted LM with twice depth as that of the base LM with roughly additional 10% more parameters.

The paper applies the proposed technique to the Pythia LM family with model sizes ranging from 160M to 1B. A collection of language understanding datasets are used for evaluation.
Compared with the corresponding base LMs, the adapted LMs based on the proposed method are observed to have at-odds performance improvements.

**Strengths:**

Enhancing/upcycling existing pretrained LMs with parameter-efficient methods to achieve new capabilities is an important research topic.
The proposed method is a reasonable technique.

**Weaknesses:**

The exposition of the paper requires substantial improvements:

*The cross layer parameter sharing is a widely studied technique in previous work (e.g., albert [1] inter alia). It is necessary to cite and discuss properly.

*Please update the citation based on ICLR recommendations, e.g., using \citep.

*Please provide proper citations for models/methods/datasets used in the paper, e..g, line 066, line 287. Without proper citations, it is hard to evaluate whether the experiment set up is properly and the comparison is meaningful.

*As the entire input token sequence is used for the second pass, it is good to reflect that in Fig 1.

The experiment setup is problematic without enough convincing evidences:

*Across all considered LMs with varying sizes and datasets, the improvements of the proposed method over base LM are at odds. Even for cases where there are certain improvements (e.g., WSC), the paper fails to include any insights, e.g., what are those improved cases and are those results statistically significant?

*As the proposed techniques trade-off the computation complexity for parameter efficiency, it is good to picture the performance vs computation costs between base LMs and the proposed method. Without, it is hard to justify whether the extra costs are truly worthy.

*It is good to test the robustness of those chosen hyperparameters. For example, It is unclear how those few-shot examples are chosen and how sensitive those decisions are. How the 1B token training dataset is selected from Pile and what domains are included?

*Although it is good to consider LMs of varying sizes, it is better to include LMs from other families. This could provide more insights on the generalizability of the proposed method, e.g., Transformer architecture variants, pretraining corpus and tokenizers.


[1] Lan et al., ALBERT: A LITE BERT FOR SELF-SUPERVISED LEARNING OF LANGUAGE REPRESENTATIONS

**Questions:**

For the second pass, is the token retrospect allowed to attend over previously generated token hiddens in the second pass? Why or why not?

Can you provide more info on the continual pretraining? What hardware is used? For the 1B token training dataset, how many epochs are used? Do you see any benefit with more tokens or longer training?

---

> ### Author Response · Authors · 2024-11-20
> **Response to Reviewer r8Jr (1/3)**
>
> Dear Reviewer r8Jr,
>
> Thank you for your valuable feedback and for taking the time to review our paper. We appreciate your insightful comments and suggestions, which have helped us improve the quality of our work. Below, we address each of your concerns in detail.
>
> **Weakness**
>
> 1. The exposition of the paper requires substantial improvements:
>
>   - **Cross layer parameter sharing:** Thank you for pointing out the need to discuss cross-layer parameter sharing. While the primary goal of the work by Lan et al. in "ALBERT: A LITE BERT FOR SELF-SUPERVISED LEARNING OF LANGUAGE REPRESENTATIONS" is to reduce memory usage, our focus is on improving performance through the stutter mechanism. We will include a citation and discussion of this relevant work in our updated paper to provide a comprehensive context for our approach.
>
>   - **Citation format:** We will update the citations in our paper to follow the ICLR recommendations, using \citep where appropriate.
>
>
>   - **Proper citations for models/methods/datasets:** We will provide proper citations for all models, methods, and datasets used in the paper, including those mentioned on line 066 and line 287. This will ensure that the experimental setup is clear and the comparisons are meaningful.
>
>   - **Figure 1 update:** We will update Figure 1 to reflect that the entire input token sequence is used for the second pass, providing a clearer representation of our method.
>
> 2. The experiment setup is problematic without enough convincing evidence:
>
>   - **Inconsistent Improvements and lack of insights**: We appreciate the reviewer's feedback regarding the varying improvements observed across different LMs and datasets. We acknowledge that while there are certain improvements, such as in the WSC benchmark, our paper does not delve deeply into the specific cases or statistical significance of these results. We are actively investigating these aspects to gain a better understanding of the improvements and their underlying causes and will provide detailed insights next week.
>
>   - **Performance vs computation costs**:
>
>     - **Training**: In our current setting in training, we stutter at every token. Our method employs a two-pass process. For a given sample ( $X = {x_1, x_2, ..., x_n}$ ):
>
>     1. **First Pass**: We store the hidden states (${h^1_l*, h^2_l*, ..., h^n_l*}$) for each token in the sequence.
>
>     2. **Second Pass (Stutter Phase)**: During this phase, we utilize the stored hidden states by passing both the stored hidden states and the current token to the model. For instance, to generate the final prediction for the next token of ( $x_3$ ), we input ( $x_2$ ) and ( $h^2_l*$ ) to the stutter model. The original Feed-Forward (FF) and Attention (Attn) mechanisms perform the same as in the base model, where ( $x_2$ ) will attend on ( $x_1$ ). Our token retrospect mechanism performs linear attention using ( $x_2$ ) and ( $h^2_l*$ ) to extract information from the first pass to generate the next token.
>
>       If the base model has time complexity O(n), the time complexity of stutter model has time complexity 2 x O(n) since we run each sample twice, remaining the same time complexity as the base model.

---

> ### Author Response · Authors · 2024-11-20
> **Response to Reviewer r8Jr (2/3)**
>
> **Weakness**
> - **Performance vs computation costs**:
>     - **Inference**: In the benchmark dataset we use for inference, the only token we use for stutter is the choice of the samples. In the Lambada-openai dataset, it only predicts the last token. In the multiple-choice dataset, the only tokens we stutter are the choice tokens. For example, in the Lambada dataset, given the context:
>       - Given: "He heard Rihanna speak 'The Queen wants you in her carriage.' Tom spoke 'No, I’m not going in some asylum.' Ran was seen standing next to him spoke 'It’s just for a private talk with you that’s all.' Tom groaned and went inside the carriage to sit down next to the"
>       - The goal is to predict "Queen"
>       - So we actually just stutter at the last token.
>
>      For another example from multiple choice dataset:
>     - Question: "George wants to warm his hands quickly by rubbing them. Which skin surface will produce the most heat?"
>     - Choice: ["dry palms", "wet palms", "palms covered with oil", "palms covered with lotion"]
>
>     The test was to concatenate the question and each answer, so there will be four inputs, and the model calculation is given by the sum of the lowest log probability of the choice part:
>
>     1. "George wants to warm his hands quickly by rubbing them. Which skin surface will produce the most heat? dry palms"
>     2. "George wants to warm his hands quickly by rubbing them. Which skin surface will produce the most heat? wet palms"
>     3. "George wants to warm his hands quickly by rubbing them. Which skin surface will produce the most heat? palms covered with oil"
>     4. "George wants to warm his hands quickly by rubbing them. Which skin surface will produce the most heat? palms covered with lotion"
>
>     For inference, we will only stutter the choice part (which is "dry palms", "wet palms", "palms covered with oil", and "palms covered with lotion".) or the last token. Therefore, the time complexity for inference with the stutter mechanism is  O(n) for the first pass and  O(1) for each stuttered token in the second pass, the overall time complexity remains O(n), similar to the base model. This ensures that the stutter mechanism does not significantly increase the computational complexity during inference.
>
> - **Hyperparameter robustness**:  We will provide more details on the selection and sensitivity of the few-shot examples and the 1B token training dataset from Pile. In our current setting. For the current 1B dataset used, it was randomly selected from the shuffled Pile dataset. The Pile is a diverse and comprehensive dataset that includes a wide range of domains such as web pages, academic papers, books, code repositories, legal documents, forums, patents, medical abstracts, literature, subtitles, encyclopedic entries, mathematical content, chat logs, and more. This variety ensures that the training data covers a broad spectrum of topics and styles, contributing to the robustness and generalizability of the model
>
> - **Inclusion of LMs from other families:** We understand the importance of evaluating our approach against prominent models. We are currently experimenting with the stutter mechanism on LLAMA 1B and plan to provide an update with detailed results and comparisons next week.

---

> ### Author Response · Authors · 2024-11-20
> **Response to Reviewer r8Jr (3/3)**
>
> **Questions**
>
> (1) For the second pass, is the token retrospect allowed to attend over previously generated token hiddens in the second pass? Why or why not?
>
> No, in our current implementation, the token retrospect is not allowed to attend over previously generated token hiddens in the second pass. We aimed to keep our mechanism simple and effective in this prototype. However, we acknowledge that allowing the token retrospect to attend over previously generated token hiddens could be a valuable enhancement. We plan to explore different kinds of attention mechanisms in future work to further improve the performance and flexibility of our approach.
>
> (2) Can you provide more info on the continual pretraining? What hardware is used? For the 1B token training dataset, how many epochs are used? Do you see any benefit with more tokens or longer training?
>
> Here are the details:
>
> - **Hardware**: We utilized 4 NVIDIA A6000 GPUs for the continual pretraining process.
>
> - **Training Dataset**: The 1B token training dataset was processed for 1 epoch.
>
> - **Hyperparameters**:
>
>   - Learning rate: 5e-5
>   - Scheduler: Cosine scheduler with warmup
>   - Warmup ratio: 0.01
>   - Optimizer: Adam
>   - Gradient accumulation steps: 8
> - **Token and Training Duration**:We observed that increasing the number of tokens to 2B provided a slight improvement in performance. However, the performance gains were marginal and did not justify the additional computational cost. Therefore, we found that training with 1B tokens strikes a good balance between efficiency and performance.
>
>
> Thank you again for your valuable feedback. We hope that the additional clarifications and improvements we have made to our paper address your concerns. We kindly request that you consider these enhancements when re-evaluating our paper and feel free to let us know if you have further questions.

---

> > ### Comment · Reviewer_r8Jr · 2024-11-22
> >
> > Thanks the authors for providing more details.
> >
> > There is no further question from my side.

---

> > > ### Author Response · Authors · 2024-12-03
> > >
> > > Dear Reviewer r9Jr,
> > >
> > > Thanks for your response! Based on your suggestions, we have made the following updates to our paper:
> > >
> > > 1. Updated the figures and descriptions to better reflect the two pass approach of mechanism.
> > > 2. Revised the citation format.
> > > 3. Included discussions and citations on cross-layer attention.
> > > 4. Add citation of the model and dataset used in the experiments, and include experimental details.
> > > 5. Discussed training and inference complexity and experimented with the stutter mechanism on Llama-1b in the appendix.
> > >
> > > Thank you again for your valuable insights.

---

### Official Review · Reviewer_fJGV · 2024-11-04

**Soundness:** 2
**Presentation:** 3
**Contribution:** 3
**Rating:** 5
**Confidence:** 5

**Summary:**

The paper proposed a new mechanism called "stutter" to allow the model to use more compute during the inference time. The mechanism works in the following way: for every token, the model encodes it twice, and the second time, each layer has a module that attends to the last step's last layer representation. This allows the model to "think twice". The design is very simple and minimalistic and requires as few as 1B tokens to train.

The proposed method has huge extensibility: future work can explore when to "stutter" (for example, at more difficult prediction steps) and can stutter for multiple steps. This is similar to the recently popular "inference scaling" scheme. With almost no increase in the model parameters, the method allows the model to use more compute in the inference time, hence better results.

The authors conducted experiments on several small-scale Pythia models, which show some improvement on standard benchmarks in both few-shot/zero-shot settings. There are also ablations on the effect of which layer that stutter modules attend to and how many times to stutter. While this is still a very initial exploration (no experiments on potential stutter location selection; no experiments on truly generative tasks), the idea is very interesting and holds a lot of potential.

**Strengths:**

(1) The proposed method is clear, intuitive, and simple. The method holds a lot of potential in the inference-scaling scheme.

(2) The empirical gain on small models are significant on certain tasks.

**Weaknesses:**

Though I really like the idea and think this can have a huge impact in future research, I have several concerns in terms of effectiveness, speed, and baseline.

(1) The selected tasks mostly don't require complex reasoning (unlike tasks like GSM8K). It is unclear how this method will perform on tasks that truly require reasoning. I understand that tasks like coding/GSM8K will have only trivial results at this scale (1B), but maybe the authors can explore some more synthetic tasks that require multi-hop reasoning.

(2) The gain is not very consistent or significant across different tasks. In this case, the authors should also report variance to show the significance of the results.

(3) **My biggest concern** is that the method is extremely inefficient in inference. The current setting is that the model stutters at every token; since each stutter step needs to look at the last step's last layer, this essentially turns the parallel prefix filling (encoding the context) into an autoregressive procedure, which will be extremely slow, especially when the prefix is long. To the best of my knowledge, the authors did not discuss this. One remedy I can think of is to only stutter at the last token before the model outputs the answer.

(4) The authors did not include discussion/comparison to a very relevant method: pause tokens (Goyal et al.). In fact, pause tokens are more efficient because they can still encode the prefix in parallel like standard transformers instead of the autoregressive style.

Goyal et al. Think before you speak: Training Language Models With Pause Tokens

**Questions:**

Please see the weakness section.

---

> ### Author Response · Authors · 2024-11-20
> **Response to Reviewer fJGV (1/2)**
>
> Dear Reviewer fJGV,
>
> Thank you for your valuable feedback and for taking the time to review our paper. We appreciate your insightful comments and suggestions, which have helped us improve the quality of our work. Below, we address each of your concerns in detail.
>
> (1) **The selected tasks mostly don't require complex reasoning**: The benchmark datasets we used are aligned with those reported in the Pythia paper, and they do include reasoning tasks. However, we acknowledge the importance of evaluating our method on more complex reasoning tasks. We are currently exploring more complex datasets like GSM8K and synthetic tasks that require multi-hop reasoning to further assess the effectiveness of our approach. We appreciate your suggestion and are committed to providing a more comprehensive evaluation in future work.
>
> (2) **The gain is not very consistent or significant across different tasks. In this case, the authors should also report variance to show the significance of the results**: To address this concern, we plan to report the variance of our results to provide a clearer picture of their statistical significance. We will measure the variance by training multiple models with different random seeds and reporting the variance of the results. This approach will help us understand the stability and reliability of the performance improvements introduced by the stutter mechanism. We are currently conducting these additional experiments and will include the variance in our updated results next week to provide a more comprehensive evaluation of our method's effectiveness.
>
> (3) **Inefficiency in inference**: In the benchmark dataset we use for inference, the only token we use for stutter is the choice of the samples. In the Lambada-openai dataset, it only predicts the last token. In the multiple-choice dataset, the only tokens we stutter are the choice tokens. For example, in the Lambada dataset, given the context:
>
> - Given: "He heard Rihanna speak 'The Queen wants you in her carriage.' Tom spoke 'No, I’m not going in some asylum.' Ran was seen standing next to him spoke 'It’s just for a private talk with you that’s all.' Tom groaned and went inside the carriage to sit down next to the"
>
> - The goal is to predict "Queen"
>
> - So we actually just stutter at the last token.
>
>
>     For another example from multiple choice dataset:
>
> - Question: "George wants to warm his hands quickly by rubbing them. Which skin surface will produce the most heat?"
>
> - Choice: ["dry palms", "wet palms", "palms covered with oil", "palms covered with lotion"]
>
>
> The test was to concatenate the question and each answer, so there will be four inputs, and the model calculation is given by the sum of the lowest log probability of the choice part:
>
> 1. "George wants to warm his hands quickly by rubbing them. Which skin surface will produce the most heat? dry palms"
>
> 2. "George wants to warm his hands quickly by rubbing them. Which skin surface will produce the most heat? wet palms"
>
> 3. "George wants to warm his hands quickly by rubbing them. Which skin surface will produce the most heat? palms covered with oil"
>
> 4. "George wants to warm his hands quickly by rubbing them. Which skin surface will produce the most heat? palms covered with lotion"
>
>
> For inference, we will only stutter the choice part (which is "dry palms", "wet palms", "palms covered with oil", and "palms covered with lotion") or the last token. Therefore, the time complexity for inference with the stutter mechanism is  O(n) for the first pass and  O(1) for each stuttered token in the second pass. The overall time complexity remains O(n), similar to the base model. This ensures that the stutter mechanism does not significantly increase the computational complexity during inference.

---

> ### Author Response · Authors · 2024-11-20
> **Response to Reviewer fJGV (2/2)**
>
> (4) **The authors did not include discussion/comparison to a very relevant method: pause tokens (Goyal et al.)**: Thank you for pointing out the relevance of the pause tokens method by Goyal et al. We appreciate the opportunity to discuss and compare our approach with theirs.
>
> While pause tokens are indeed an efficient method that allows encoding the prefix in parallel, there are some key differences and advantages to our stutter mechanism:
>
> 1. **Efficiency in Pretraining**: The pause tokens method involves pretraining on the entire C4 dataset, which requires substantial computational resources and time. In contrast, our stutter mechanism is designed to be more efficient, requiring only 1 billion tokens for pretraining. This makes our approach more accessible and less resource-intensive.
>
> 2. **Parameter Updates**: The pause tokens method updates all parameters during training, which can be computationally expensive. Our stutter mechanism, on the other hand, updates only a small subset of parameters, making it more efficient in terms of both computation and memory usage.
>
> 3. **Pause tokens**: The pause tokens method introduces special tokens into the input sequence, which Goyal et al. themselves noted may not provide meaningful information to the model. In contrast, our stutter mechanism uses the hidden state of the previous token, allowing the model to "think again" and continue reasoning from the previous context. This approach provides a more coherent and continuous reasoning process, enhancing the model's ability to utilize prior information effectively.
>
> We acknowledge the strengths of the pause tokens method and will include a discussion and citation of this relevant work in our updated paper. We believe that our approach offers a complementary and efficient alternative, particularly for scenarios with limited computational resources.
>
> Thank you again for your valuable feedback. We hope that the additional clarifications and improvements we have made to our paper address your concerns. We kindly request that you consider these enhancements when re-evaluating our paper and feel free to let us know if you have further questions.

---

> > ### Comment · Reviewer_fJGV · 2024-11-25
> > **Thanks for your response**
> >
> > Thanks for you response, though I am now actually confused about the method (regarding the efficiency part).
> >
> > Let's say for training, you have a sentence "This is a cat". I'll use "This1 This2 is1 is2 a1 a2 cat1 cat2" to represent the first and the shutter tokens. I understand that is2 does not see is1 except is1's last layer. My question is, does a1 see is2? Do you just do a forward pass over "This1 is1 a1 cat1" first, and then do the shutter tokens?
> >
> > If that is the case, then I get that the inference efficiency is not that bad. The authors should improve the paper to make the mechanism clearer.

---

> ### Author Response · Authors · 2024-12-03
>
> Dear Reviewer fJGV,
>
> We have updated the paper's figures and descriptions to better reflect the mechanism. Additionally, we have included a discussion and references of the pause token (Goyal et al.) in the updated version. We also include the time complexity of training and inference in the appendix. Thank you again for your valuable insights.

---

### Meta-Review · Area_Chair_hN52 · 2024-12-13

**Metareview:**

The paper introduces a novel mechanism called "stutter," designed to enable models to utilize more computational resources during inference. This mechanism operates by encoding each token twice, with the second encoding incorporating a module that attends to the last layer's representation from the previous step, effectively allowing the model to "think twice." The design is straightforward and minimalistic, requiring only about 1 billion tokens for training.

The method offers significant extensibility, as future research can investigate optimal moments to implement "stuttering" (such as during more challenging prediction steps) and potentially apply it over multiple steps. This approach aligns with the emerging trend of "inference scaling." Remarkably, it achieves improved performance without a substantial increase in model parameters.

The authors conducted experiments using several small-scale Pythia models, demonstrating improvements across standard benchmarks in both few-shot and zero-shot settings. They also performed ablation studies to assess the impact of different layers attended by the stutter modules and the frequency of stuttering. While this initial exploration lacks experiments on selecting stutter locations and does not cover truly generative tasks, the concept is intriguing and holds considerable potential.

However, the reviewers raise concerns on this paper, e.g. limited evaluation, inconsistent gain, inefficiency in inference. The authors should revise the paper as suggested by the reviewers.

**Additional Comments On Reviewer Discussion:**

However, the reviewers raise concerns on this paper, e.g. limited evaluation, inconsistent gain, inefficiency in inference. The authors should revise the paper as suggested by the reviewers.

---

### Decision · Program_Chairs · 2025-01-22

Reject